# Productivity, Satisfaction, Work Environment and Health after Relocation to an Activity-Based Flex Office—The Active Office Design Study

**DOI:** 10.3390/ijerph18147640

**Published:** 2021-07-18

**Authors:** Maria Öhrn, Viktoria Wahlström, Mette S. Harder, Maria Nordin, Anita Pettersson-Strömbäck, Christina Bodin Danielsson, David Olsson, Martin Andersson, Lisbeth Slunga Järvholm

**Affiliations:** 1Department of Public Health and Clinical Medicine, Section of Sustainable Health, Umeå University, 901 87 Umeå, Sweden; viktoria.wahlstrom@umu.se (V.W.); david.olsson@umu.se (D.O.); martin.andersson@umu.se (M.A.); lisbeth.slunga-jarvholm@umu.se (L.S.J.); 2Umeå School of Architecture, Umeå University, 901 87 Umeå, Sweden; mette.harder@umu.se; 3Department of Psychology, Umeå University, 901 87 Umeå, Sweden; maria.nordin@umu.se (M.N.); anita.pettersson-stromback@umu.se (A.P.-S.); 4The Royal Institute of Technology (KTH), School of Architecture and the Built Environment, 100 44 Stockholm, Sweden; chrdan@kth.se

**Keywords:** activity-based work, job performance, longitudinal study, new ways of working, occupational health, office worker

## Abstract

Implementation of activity-based flex offices (AFOs) are becoming increasingly common. The aim of this study was to evaluate the effects of an AFO on perceived productivity, satisfaction, work environment and health. Questionnaire data from the longitudinal, quasi-experimental Active Office Design Study was used. The study evaluates a public organization relocating staff to either an AFO or to cell offices. Measures from baseline, 6 and 18 months after relocation, were analyzed. Employees in the AFO experienced a decreased productivity and satisfaction with the office design. Lack of privacy as well as increased noise disturbance, less satisfaction with sit comfort and work posture were reported. Employees in the AFO with work tasks requiring a high degree of concentration experienced lower productivity while those with a high proportion of teamwork rated productivity to be continually high. No significant group differences were found between the two office types in general health, cognitive stress, salutogenic health indicators or pain in the neck, shoulder or back. The study highlights the importance of taking work characteristics into account in the planning and implementation process of an AFO. Flexible and interactive tasks seem more appropriate in an AFO, whereas individual tasks demanding concentration seem less fit.

## 1. Introduction

In recent decades, work in office environment has become more common [1,2]. The development has introduced many flexible office solutions [3]. In activity-based flex offices (AFOs), employees have no assigned workstations, but choose a workplace suited for the task to be performed. In AFOs activity-based working (ABW) is applied which includes an almost entirely paperless work environment. New and better information and communication technology (ICT) solutions has made this possible. The AFO is usually dimensioned to house fewer than 70% of the employees [4] and is thus suggested to reduce facility costs. Furthermore, it is promoted for increased flexibility, more social interaction and higher work satisfaction [5,6].

Increase in productivity and work performance are aspects often highlighted when promoting AFO designs [7,8], but research is still sparse in these areas and findings are inconsistent. In a systematic review on AFO interventions, the majority of the studies included reported positive outcomes on perceived productivity and work performance [9], while other studies have shown a decrease in perceived productivity, especially when moving from cell offices to an AFO [10,11]. A study among workers in open plan offices evaluating impact of office environment on perceived productivity found differences between gender and age groups [12]. To our knowledge there are no studies performed specifically in AFOs on productivity in relation to age and gender.

Improved satisfaction with aspects of the physical work environment after relocation to an activity-based office environment have been shown in some studies, whereas other have reported negative outcomes such as increased noise distraction and lack of privacy [9,13,14]. Moreover, difficulties with concentration in AFOs have been reported, especially when comparing to cell offices [9].

Different work tasks may be better suited to different types of office. Seddigh et al. found for example that work tasks requiring a high degree of concentration were less suitable for open office designs [1]. In contrast, studies have shown a positive impact on communication and collaboration in AFOs, indicating that such work characteristics may be better supported by more flexible office settings [9]. However, employees’ satisfaction with communication, as well as privacy, also affect productivity in AFOs [15]. In a recent study, Hoendervanger et al. [16], showed that it seems to be particularly important to optimize the workers perceived fit and user behavior, to facilitate and stimulate individual high-concentration work. Interestingly, no longitudinal studies have investigated if and how productivity in AFOs is influenced depending on the actual work tasks.

Research on the impact of AFOs on health is inconsistent and thus evidence is limited [9]. In a cross-sectional study the best health and highest job satisfaction were seen among employees in cell- and flex-offices [17]. However, Meijer et al. [18] found a significant improvement in self-rated general health, whereas health was reported to be significantly lower after relocating from fixed workstations to an AFO-like environment in a controlled intervention study [19].

Many studies evaluating AFOs have been criticized for lacking a thorough description of the working conditions and how the workplace is designed to support ABW [9], and only a few studies have investigated the effects on perceived productivity, satisfaction, work environment and health when moving from a cell office to an AFO with a longitudinal design. In addition, most studies have been conducted in organizations operating in the private sector [9]. Since AFOs are becoming increasingly popular and introduced at a large scale also in the public sector, there is an urgent need to gain knowledge from this sector as well, especially as the effects on productivity and health may differ [9,20]. Moreover, it is essential to further investigate how different work tasks may influence productivity in AFOs.

The primary aim of this study was to investigate effects on perceived productivity, satisfaction, work environment and health in a group of municipal officials moving from cell offices to an AFO, and to compare with colleagues who moved from cell offices to other cell offices. The secondary aim was to investigate possible impacts of gender, age and type of work tasks on productivity in the AFO. 

We hypothesized that (1) there is no difference in perceived productivity, satisfaction, work environment and health between the two office groups over time, (2) in the AFO, employees with work tasks requiring high degree of concentration will experience a lower productivity and (3) employees with work tasks requiring a high level of collaboration and communication will experience an increased productivity in the AFO. The research question and hypotheses were investigated by using data from the Active Office Design Study (AOD Study) and by applying longitudinal data analysis to compare the results from AFOs and cell offices.

## 2. Materials and Methods

### 2.1. Design and Settings

The AOD Study is a longitudinal, non-randomized, quasi-experimental study with a reference group. During 2015, white-collar workers in a medium-sized Swedish municipality (approx. 56,000 inhabitants) relocated to new office environments. Out of 374 employees, approximately 40% relocated to new cell offices and 60% to an AFO. The majority of the participants in the study had individual workstations in single cell offices or shared rooms prior to the relocation. The group that moved to new cell offices primarily worked in social services (welfare office and social workers), while the group that moved to the AFO included employees in departments of education, urban planning, economy and human resources. In addition, politicians relocated to the AFO. The relocation to the different office settings was predetermined by the employer and both office groups relocated to renovated office buildings. The cell office building was organized on three floors with mainly single cell offices and some shared rooms. The AFO consisted of three floors, with the open plan office areas located on the 2nd and 3rd floor, equipped with workstations, as well as supporting areas including cell offices, sofas, group-and touchdown tables and different-sized meeting rooms. Clean-desk policy as well as new ICT solutions were implemented in both office buildings. For more detailed information, see Appendix A, Table A1. During the project period, there was a marked influx of refugees in Sweden, resulting in an increase in staff density in both groups and a high people to workstation ratio in the AFO.

All employees involved in the relocation were invited to participate in the AOD Study by answering a questionnaire asking about background characteristics, perceived productivity, satisfaction with the office design, work environment and health. The study included one baseline measurement 6 months prior to relocation, and two follow-up measurements, 6 and 18 months after relocation. The questionnaires were distributed at the workplace to ensure a high response rate and written informed consent was obtained from all participants prior to participation. Ethical approval was received from The Regional Ethical Committee in Sweden (No: 2104/226-31).

Due to the staff turnover as well as possible absence at any given time (i.e., due to parental leave, leave of absence, sick leave, retirement etc.) the study participants were not identical at the different measurement occasions. To achieve as accurate results as possible and to be able to do longitudinal analyses (follow the same individuals over time), only individuals with baseline data and at least one follow-up measurement was included in the analyses. For more detailed information on the study population in total and at different timepoints, see Figure 1.

### 2.2. Employee and Work Characteristics

Data on gender, age, occupation, type and degree of employment, managerial position and office type before relocation were collected at baseline.

### 2.3. Work Tasks 

The amount of computer work was registered at baseline and at follow-ups. For the 18-month follow-up, the participants were also asked to estimate the amount of h/week spent on different types of work tasks such as individual concentration intensive work tasks, teamwork in large groups (>3 people), individual routine work mainly consisting of routine tasks and amount of work involving talking on the phone. The participants also rated the importance of spontaneous meetings for their work by using a 5-point Likert scale, with a low value indicating spontaneous meetings being of great importance.

### 2.4. Outcome Variables

#### 2.4.1. Productivity and Satisfaction

Productivity was assessed by a questionnaire measuring perceived productivity and focuses on the employee’s experience of performance in relation to environmental conditions and premises at the workplace [21]. It was translated from English to Swedish by using a forward-and back translation [22] and consists of 20 statements, such as “I am able to be productive in my present workspace”, rated on a 5-point Likert scale. In the original article, the productivity scale was divided into subscales, all of which proved satisfactory internal consistency with Cronbach’s alphas ranging between 0.67 and 0.89 [21]. For the present purposes, a global scale including all the subscales was used. The mean value of all items was used as outcome measure, a higher score indicating higher productivity. Satisfaction with the office was rated by answering the question:” How satisfied are you with the design of your current office?” on a 5-point Likert scale ranging from 1 to 5 where 1 = “Very dissatisfied” and 5 = “Very satisfied”.

#### 2.4.2. Psychosocial Work Environment

The Work Experience Measurement Scale (WEMS) [23], was used to assess the psychosocial work experience. It is comprised of 32 questions divided into the six dimensions; (1) supportive working conditions (e.g., “We encourage and support each other at my workplace”), (2) internal work experience (e.g., “The work I perform is meaningful”), (3) autonomy (e.g., “I decide for myself how my work should be done”), (4) time experience (e.g., “I can keep up with my work tasks during regular working hours”), (5) management (e.g., “The manager is available when I need him/her”) and (6) reorganization (e.g., “I received relevant information concerning the latest reorganization”). The questions were rated on a 6-point Likert scale and indexes were calculated and standardized (range 0–100). A high score indicates a positive response. Cronbach’s alphas ranged between 0.75 and 0.89 for the subdimensions, respectively [23], thus indicating satisfactory internal consistency.

#### 2.4.3. Physical Work Environment

Perception of the physical environment was assessed by using validated questions concerning privacy, noise disturbance, sit comfort and work posture [24]. Noise disturbance was rated by using a 5-point Likert scale (1 = “Not disturbed, 5 = “Very disturbed”) where a high score indicated a high level of disturbance. The participants rated sit comfort and work posture (1 = “Very good”, 4 = “Very bad”) and lack of privacy (1 = “Not at all”, 4 = “To a great extent”) by using 4-point Likert scale. A high score indicated a poorer work posture and sit comfort as well as a higher degree of lack of privacy.

#### 2.4.4. Health

General health was assessed using one item (i.e., “In general, would you say your health is …”) from the SF-36 instrument [25]. The answer was rated on a 5-point Likert scale (1 = “Poor”, 5 = “Excellent”), a high score indicating a high self-rated general health. The Salutogenic Health Indicator Scale (SHIS) was used to assess well-being from a salutogenic perspective [26]. SHIS consists of 12 items including seven statements concerning well-being and five statements on how you operate and interact in relation to environmental demands. The items were rated on a 6-point Likert scale, summarized into a “total SHIS index” and standardized to a scale from 0 to 100 where a high value indicates a high degree of perceived health. According to Bringsén et al. [26], Cronbach’s alpha was 0.92 and thus satisfactory. 

Cognitive stress was assessed by using four questions from the Copenhagen Psychosocial Questionnaire (COPSOQ) [27]. On a 5-point Likert scale, participants were asked to rate how often, during the last month, they had had (1) difficulty concentrating, (2) difficulty making decisions, (3) trouble remembering and (4) difficulty thinking clearly. The mean value was used as outcome measure, a high score indicating a high level of cognitive stress. The internal consistency of the cognitive stress scale from COPSOQ has been good with Cronbach’s alpha reported to be 0.85 [27]. To assess physical discomfort, the participants rated occurrence of pain or discomfort from the neck/shoulders and back over the last three months using a 5-point Likert scale (1 = “Never”, 5 = “Always”); a higher score indicating a higher occurrence of discomfort [28].

### 2.5. Statistical Analyses

Participants with baseline data and data from at least one follow-up measurement were included in the analyses. Differences in baseline characteristics between the groups (cell office vs AFO) were investigated by using independent samples t-test and Pearson’s Chi-square tests. For continuous variables (productivity, WEMS, SHIS and COPSOQ) linear mixed models were used to examine significant interaction effects and differences between groups. The models were set up with group (two levels: AFO and cell office), time (three levels; baseline, 6 months and 18 months) and interaction (group x time) as fixed factors. Random intercepts were used for the participants in all models. Model parameters were estimated through restricted maximum likelihood (REML). Within-group effects were analyzed using the estimated marginal means of the fitted models. In a similar way, linear mixed models were set up to examine secondary outcomes, i.e., factors affecting the productivity in the AFO group, with difference over time presented as an interaction term. Some outcome variables (satisfaction, noise disturbance, sit comfort, work posture, lack of privacy, neck/shoulder/back strain, general health) were based on one-item questions and did not fulfill the requirements for using parametric statistical testing. They were therefore converted into binary variables and then analyzed using Generalized Estimating Equations (GEE) resulting in odds ratios (OR). All analyses were adjusted for age and alpha was set to 0.05. Data were processed and analyzed using the Statistical Package for the Social Sciences (SPSS; version 25; IBM Corp, Armonk, NY, USA).

## 3. Results

### 3.1. Baseline Characteristics

At baseline, 336 (91.3%) individuals answered the questionnaire. Out of these, 287 participants with baseline data and data from at least one follow-up measurement, 190 in the AFO group and 97 in the cell office group, were included in the analyses. The AFO group consisted of 61 men and 129 women and the cell office group of 5 men and 92 women. The mean age was 47.7 and 44.7, respectively. Baseline data are presented in Table 1. Statistically significant differences between the office groups were seen for age (*p* < 0.01), sex (*p* < 0.001), degree of employment (*p* < 0.01), proportion of managers (*p* < 0.001) and office type before relocation (*p* < 0.001). No significant difference was seen regarding self-rated general health.

### 3.2. Productivity and Satisfaction

Group estimated means and odds ratios for perceived productivity and satisfaction, respectively, are displayed in Table 2. There was a statistically significant group over time effect in productivity (*p <* 0.001), where the employees in the AFO rated a decline in productivity after relocation. Within group analysis showed a significant decrease in productivity in the AFO group at 6 months (*p <* 0.001), as well as at 18 months (*p <* 0.001) after relocation. There was no difference in productivity within the cell office at different time points. Results from the GEE analyses showed a significant group over time effect regarding satisfaction with the office design (*p <* 0.001). Employees who moved to the AFO rated a higher satisfaction at baseline compared to the employees moving to the cell office (OR *=* 5.32) but displayed a reduction in satisfaction with the office design at 6 months (OR *=* 1.12) and 18 months (OR *=* 0.86). Employees working in the cell offices on the other hand, reported increased satisfaction after relocation (OR 6 months *=* 2.96, OR 18 months = 4.86).

### 3.3. Psychosocial Work Environment

There was a significant group over time effect in the dimension supportive working conditions in the WEMS questionnaire (*p <* 0.01) (Table 2), with a decrease in rating within the AFO group at 6 months (*p <* 0.001), while no changes over time were seen over time in the cell office group. There was no group over time effect regarding other WEMS dimensions. However, within group analyses showed a significant decrease in rating of the dimensions internal work experience (6 months; *p* < 0.001, 18 months; *p <* 0.01) and management (6 months; *p <* 0.05) in the AFO group. In the cell office group, there was a significant increase in rating of autonomy at 18 months (*p <* 0.05).

### 3.4. Physical Work Environment

As displayed in Table 3 there were statistically significant group over time effects regarding the physical aspects of the work environment, i.e., noise disturbance (*p <* 0.001), lack of privacy (*p <* 0.001), sit comfort (*p <* 0.01) and work posture (*p =* 0.001). At baseline, the group that moved to the AFO rated less disturbance from noise (OR *=* 0.22), and lack of privacy (OR *=* 0.40) as well as better sit comfort (OR *=* 4.04) and better work posture (OR *=* 4.61), compared to the cell office group. After relocation, ratings regarding noise disturbance and lack of privacy deteriorated in the AFO group at both follow-ups. In addition, the odds of reporting a good sit comfort and work posture declined in the AFO group after relocation. In the cell office on the other hand, ratings in noise disturbance and lack of privacy were lower after relocation. In contrast to the AFO group, the odds of reporting good sit comfort and work posture increased in the cell office group after relocation.

### 3.5. Health

No significant group over time effects were found regarding self-rated general health, salutogenic health indicators (SHIS), cognitive stress (COPSOQ) or discomfort in the neck, shoulders or back (Table 3). Cognitive stress significantly increased in the AFO group at the 6-month follow-up (*p <* 0.01), but not at 18 months. Employees in the cell office group rated a significant decrease in the SHIS-score and a higher degree of cognitive stress at both follow ups (*p <* 0.05).

### 3.6. Productivity in the AFO

When analyzing data from the employees in the AFO, there was a significant interaction effect in hours per week spent on individual work tasks requiring concentration and productivity (*p <* 0.001), indicating that productivity decreased with increased time spent on these tasks (Table 4, Figure 2). A significant interaction effect was also found for productivity in relation to the amount of teamwork performed (*p =* 0.011). More time spent working in teams was associated with a higher estimated productivity after relocation (Figure 2). The importance of spontaneous meetings also interacted with productivity (*p <* 0.001). A low degree of importance was associated with a lower productivity (Figure 2). A significant interaction effect was seen between productivity and being a manager or not (*p <* 0.001). In the non-manager group productivity decreased after relocation compared to the manager group (Figure 2). Further analyses of work tasks showed that managers, compared to non-managers, worked from home more (*p <* 0.001), had more meetings outside of the office (*p =* 0.04), worked more in groups (*p <* 0.001), had a smaller proportion of individual tasks requiring concentration (*p =* 0.001) and rated spontaneous meetings to be more important for their work (*p* < 0.001). No significant interaction effects were seen for productivity in relation to age, gender, time spent on performing routine tasks, computer work or working over the phone.

## 4. Discussion

The primary aim of this study was to investigate the effect on employees’ perceived productivity, satisfaction with the office design, work environment and health when relocating from cell offices to an AFO in comparison to a group in the same organization relocating from cell offices to other cell offices.

We hypothesized that there would be no significant difference in development of productivity or satisfaction between the two office groups after relocation. Our results did not support this as employees in the AFO group reported a decrease in perceived productivity while the cell office group’s productivity rating remained unchanged. This finding is in contrast to earlier studies that have shown improved productivity in activity based and flexible office environments [6,18,29,30].

Our study is unique, as we could analyze the impact of work tasks on productivity in an AFO by using longitudinal data. The results supported our other hypotheses since employees with individual concentration intensive work tasks experienced a lower productivity in the AFO. We also found that employees with work tasks requiring a lot of collaboration and communication experienced an unchanged high level of productivity in the AFO. Productivity was rated higher among employees in the AFO who considered spontaneous meetings to be of high importance, as well as for those who spent more time working in teams. These results are in line with previous research were ABW has been shown to be unfavorable for concentration and privacy but positively associated with areas of communication and collaboration [9]. Furthermore, we found that employees who worked in the AFO but had no managerial position experienced a lower productivity after relocation compared to managers. This can possibly be explained by managers performing more mobile and communicative tasks. The fit between specific work patterns (i.e., interactivity) and the office environment has shown to be of great importance in relation to employee performance [31]. It appears as though flexible and interactive work tasks are appropriate for an AFO environment whereas individual tasks demanding a high degree of concentration are less suitable. Furthermore, the facilitation and access to quiet areas and closed work settings for concentration intensive work seems to be of great importance as has been highlighted in previous studies [16,32]. Another factor that has been associated with possible impact on productivity in AFOs, is the time that employees spend searching for a workstation. Haapakangas et al. [15] found that increased time spent on finding a suitable workspace was associated with lower productivity. It is not unlikely that the high employee density that occurred during the course of our study affected the access to available workstations in the AFO negatively, which in turn could have impaired productivity ratings. We found no differences in perceived productivity in the AFO in relation to age or gender. This could be due to work tasks being homogeneous between older and younger employees as well as between men and women.

Our results showed that satisfaction with the office design in the AFO group significantly decreased, whereas in the cell office group the satisfaction increased after relocation. This is in contrast to previous research that showed higher satisfaction in activity based and flexible office settings [13,30]. The need for privacy has been shown to play an important role in employees’ satisfaction with the work environment inherent to AFOs [14]. In addition, satisfaction with possibilities of privacy and communication is strongly associated with positive productivity outcome in the AFOs [15]. The unfavorable influences from the physical work environment can be a possible explanation of the decrease in both satisfaction and productivity in the AFO group. These results are consistent with previous research showing that employees in AFOs reported decreased satisfaction with privacy compared to employees working in individual offices [11]. Moreover, the high occupancy rate at the time of the AOD Study may have had a negative impact on both noise levels and the opportunity to work in privacy.

Another factor that could influence the ratings of productivity and satisfaction is the type of office used prior to relocation. In our study the employees were mainly relocated from cell offices. However, in many of the previous studies, the employees worked in open plan offices with individual workstations before relocating. Open office landscapes have been associated with higher levels of cognitive stress and distraction [1] and negative effects on perceived work environment [4]. It is thus possible that a relocation from an open plan office to an AFO would increase satisfaction and perceived productivity, whereas relocating from cell offices to an AFO might result in a decrease in satisfaction and productivity.

Interestingly, our results showed a significant difference in experience of supportive working conditions between the groups over time. A decline in perceived support was seen within the AFO group at the 6-month follow-up. The internal work experience as well as perception of management also declined within the AFO group after relocation. This is in line with the results of Morrison et al. [33], who found that perceived supervisory support decreases in shared working environments, especially when employees have no assigned workstations. Managers need to adjust their leadership behavior and find new ways to secure the sharing of information and team coherence in AFOs. In addition, an adaptation to new ways of communication seems necessary since initial adverse effects on communication between managers and employees have been reported [34]. Thus, our results may reflect the need for reorientation regarding organization, working methods and leadership strategies.

Experiences regarding the physical work environment also differed between the groups in the present study. Employees in the AFO experienced a decline in sit comfort and work posture, indicating difficulties achieving optimal ergonomic conditions. Ergonomic adjustments in AFOs can be perceived as both difficult and time consuming [35], and adverse effects in employees’ perceived productivity and health has been found when satisfaction and adjustability with furniture comfort and workspace decreased [36]. With the basic idea of switching workplaces in the AFO, employees may have to adjust their workstation several times a day. It is therefore important to acknowledge the new ergonomic challenges of AFOs.

As we hypothesized, there were no significant differences in any of the health aspects between the office groups over time. Neither were there any differences between the groups regarding pain or discomfort in the neck, shoulder or back. However, there seemed to be a downward trend in both groups, with the probability of reporting good health declining over time. This is in contrast to what Meijer et al. [18] found in a longitudinal study within the public sector, where general health increased significantly at follow-ups and upper extremity complaints decreased, for employees moving from cell offices to an AFO-like environment. However, that study population comprised employees with mainly computer work, who possibly had more homogenous work tasks than the participants in our study. Moreover, Nijp et al. [19] found a decrease in general health in a group working in an ABW environment, but no change in a reference group.

The workload and crowding increased in general over time in our sample. This affected both office groups, possibly explaining the similar decrease in general health and increase in cognitive stress. Moreover, it is possible that adverse effects in general health and stress can be seen first after an even longer exposure time [1]. As there seems to be an increased risk of sickness absence in open-plan workspaces compared to cell offices [37], it is unfortunate that we did not have the possibility to follow sick-leave rates in our study. More research is needed to investigate this further.

### Strengths and Limitations

One of the strengths of this study is the controlled, longitudinal design and the long follow-up time which enabled us to study both short and long-term effects. The high response rate should ensure reliable and representative data of the employees involved in the office relocation. Furthermore, we used validated instruments with good internal consistency, to measure productivity, work environment and health. Our study also presents novel results on how different work task characteristics affect productivity in an AFO.

The study also had some limitations. Since the employee allocation had been predetermined by the employer, randomization was not possible. Therefore, there were anticipated differences between the two office groups due to different assignments, e.g., in gender and managerial status, which could have had an impact on the results. However, through the use of a reference group, changes over time that are likely to have affected both groups in similar ways could be disregarded. The fact that the whole study was performed in the same large organization is probably an advantage. Still there were some important differences between the two groups that we could not take into account which probably reduces the generalizability of the results.

Some selection bias cannot be excluded. It is not unlikely that employees who were dissatisfied with the office design or experienced negative health effects in relation to the office, quit their work and were therefore not included in the study.

## 5. Conclusions

This study shows that perceived productivity as well as satisfaction with the office design decreased when employees within a municipality in Sweden relocated from cell offices to an AFO. Productivity was strongly related to work characteristics which highlights the importance of taking the type of assignments, work tasks and organizational factors into account when planning for and implementing an AFO. Moreover, lack of privacy, as well as noise disturbance, increased after relocation. This seems to be a general problem occurring in AFOs that needs to be addressed already in the planning stage by, e.g., adding enough separate and quiet settings. Experiences of reduced sit comfort and impaired work posture illuminate the potential ergonomic challenges in an AFO. We found no change in general health, but research regarding long-term health effects in AFOs is sparse and more studies addressing this issue, as well as sickness absence data, is needed to draw well-grounded conclusions. Lastly, our study indicates that one size does not fit all, and that there is an urgent need for studies to increase the knowledge about how AFOs should be planned and adapted to suit employees with different kinds of work tasks.

## Figures and Tables

**Figure 1 ijerph-18-07640-f001:**
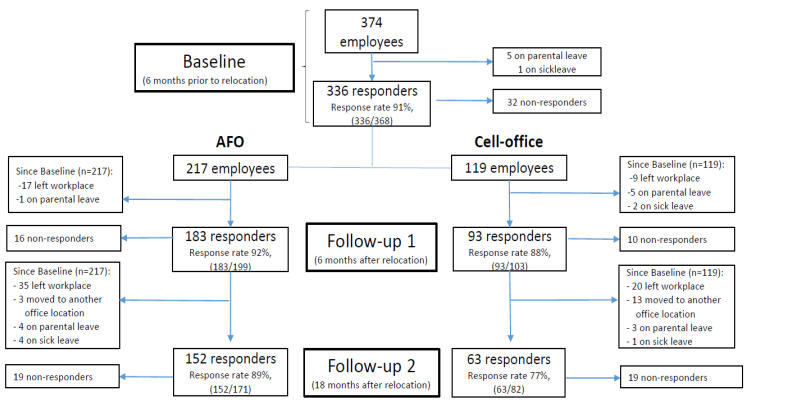
Flowchart of the original study population in total and at different time-points, including drop-outs.

**Figure 2 ijerph-18-07640-f002:**
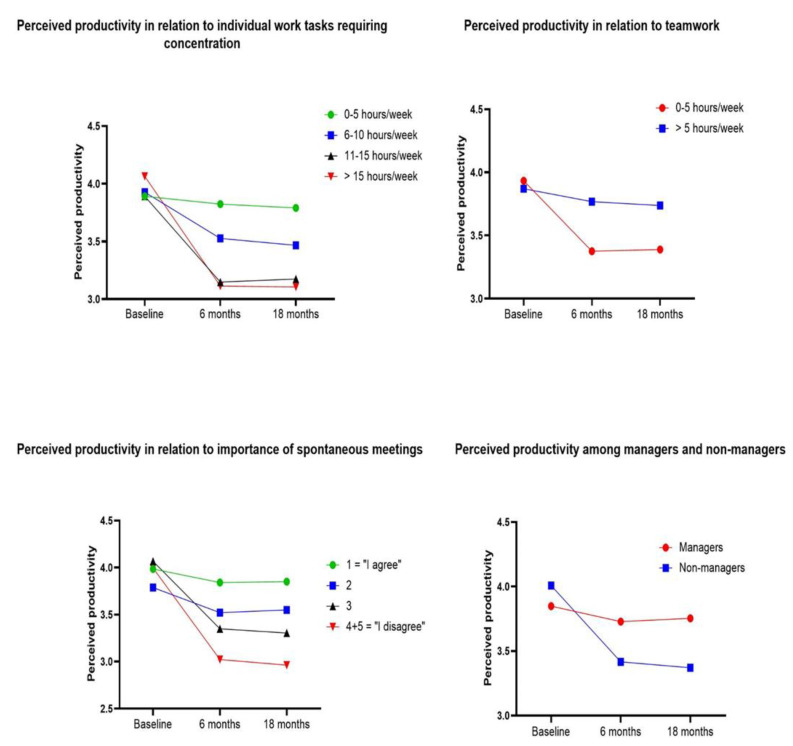
Perceived productivity (range 0–5, a high value indicating a high productivity) in the AFO in relation to time spent on individual work tasks requiring concentration (h/week), time spent working in teams (h/week), importance of spontaneous meetings (scale 1–5, 1 = “I agree”, 5 = “I disagree”. A low value indicating a high importance of spontaneous meetings), and among managers/non-managers. Presented as estimated means (EM). Significant time interaction effects were found for number of hours spent on individual work tasks requiring concentration (*p <* 0.001), number of hours working in teams (*p =* 0.011), importance of spontaneous meetings (*p <* 0.001) and among managers/non-managers (*p <* 0.001).

**Table 1 ijerph-18-07640-t001:** Baseline descriptions of participants in the activity-based flex office (AFO) and cell office group, respectively.

	AFO (n = 190)		Cell Office (n = 97)	
	Mean (SD)	n (%)	Mean (SD)	n (%)
**Age ^1^**	47.7 (10.3)		44.7 (11.1)	
**Sex**				
Female		129 (68)		92 (95)
Male		61 (32)		5 (5)
**Degree of Employment**				
100%		177 (93.2)		77 (79.4)
75–99%		12 (6.3)		17 (17.5)
40–74%		1 (0.5)		2 (2.1)
Missing		-		1 (1.0)
**Manager**				
Yes		51 (27)		3 (3)
No		139 (73)		93 (96)
Missing		-		1 (1)
**Office Type Before Relocation**				
Cell office (1 person)		126 (66)		56 (58)
Shared room (2–3 persons)		20 (11)		39 (40)
Open plan office		23 (12)		0 (0)
No assigned workspace		17 (9)		1 (1)
Missing		4 (2)		1 (1)
**Self-Rated General Health**				
Excellent		28 (15)		9 (9)
Very good		88 (46)		39 (40)
Good		56 (30)		32 (33)
Moderate		15 (8)		17 (18)
Poor		2 (1)		0 (0)
Missing		1 (0.5)		0 (0)
Missing				

^1^ Missing data from two participants.

**Table 2 ijerph-18-07640-t002:** Comparisons of perceived productivity, satisfaction and psychosocial working conditions (WEMS, Work Experience Measurement Scale) in the activity-based flex office (AFO) and cell office, expressed as estimated means (EM) and odds ratios (OR). Difference between the groups over time is expressed as group x time effect. For estimated means (EM), pairwise comparisons within groups are presented. Bold indicates statistically significant within group differences.

	AFO			Cell Office			*p*-Value for Group x Time
	EM	OR	95% CI	EM	OR	95% CI	Effect
**Productivity**							<0.001
Baseline	3.97		3.89–4.04	3.50		3.40–3.60	
6 months	**3.50 ^a^**		3.41–3.58	3.58		3.46–3.70	
18 months	**3.47 ^a^**		3.38–3.55	3.54		3.41–3.67	
**Satisfaction with the Office Design**							<0.001
Baseline		5.32	2.48–11.42		1 ^d^		
6 months		1.12	0.63–2.00		2.96	1.26–7.11	
18 months		0.86	0.48–1.54		4.86	1.59–14.8	
**WEMS** ***Supportive Working Condition***							0.006
Baseline	76.9		74.4–79.3	71.4		67.9–74.8	
6 months	**72.0 ^a^**		**69.3–74.6**	72.3		68.8–76.1	
18 months	74.5		71.9–77.2	70.1		66.2–74.0	
***Internal Work Experience***							0.07
Baseline	81.6		79.5–83.8	78.1		75.0–81.2	
6 months	**77.3 ^a^**		**74.8–79.7**	77.1		73.7–80.6	
18 months	**77.9 ^b^**		**75.1–80.8**	75.1		70.9–79.3	
***Autonomy***							0.10
Baseline	68.3		65.6–70.9	53.6		49.8–57.4	
6 months	68.4		65.7–71.0	56.2		52.5–59.9	
18 months	68.5		65.7–71.2	**59.1^c^**		**54.9–63.3**	
***Time Experience***							0.09
Baseline	47.8		44.4–51.2	41.9		37.2–46.6	
6 months	48.2		44.7–51.7	41.3		36.4–46.3	
18 months	48.6		45.1–52.2	48.3		43.1–53.6	
***Management***							0.40
Baseline	72.0		69.1–74.8	63.4		59.5–67.4	
6 months	**68.2 ^c^**		**65.0–71.4**	62.9		58.4–67.3	
18 months	70.1		66.7–73.4	64.1		59.0–69.2	
***Reorganization***							0.54
Baseline	60.0		56.4–63.5	60.8		55.8–65.8	
6 months	57.9		54.1–61.7	61.6		56.3–67.0	
18 months	59.4		55.2–63.5	64.0		57.8–70.3	

^a^ *p*-value < 0.001, ^b^ *p*-value *<* 0.01, ^c^ *p*-value *<* 0.05, ^d^ reference.

**Table 3 ijerph-18-07640-t003:** Comparisons of physical working conditions and health in the activity-based flex office (AFO) and cell office, expressed as estimated means (EM) and odds ratios (OR). Difference between the groups over time is expressed as group x time effect. For estimated means (EM), pairwise comparisons within groups are presented. Bold indicates statistically significant within group differences.

	AFO			Cell Office			*p*-Value for Group x Time
	EM	OR	95% CI	EM	OR	95% CI	
**Disturbance from Noise (Voices etc.)**							<0.001
Baseline		0.22	0.12–0.38		1 ^c^		
6 months		0.79	0.48–1.29		0.73	0.44–1.22	
18 months		1.05	0.63–1.76		0.62	0.33–1.18	
**Lack of Privacy**							<0.001
Baseline		0.40	0.21–0.75		1 ^c^		
6 months		2.85	1.66–4.87		0.47	0.23–0.998	
18 months		3.10	1.77–5.37		0.61	0.30–1.24	
**Sit Comfort**							0.005
Baseline		4.04	2.04–7.98		1 ^c^		
6 months		1.28	0.72–2.26		1.13	0.68–1.89	
18 months		1.39	0.77–2.53		1.30	0.69–2.46	
**Work Posture**							0.001
Baseline		4.61	1.89–11.21		1 ^c^		
6 months		0.85	0.44–1.62		1.35	0.67–2.72	
18 months		0.92	0.46–1.81		1.59	0.64–3.95	
**Self-Rated General Health**							0.78
Baseline		2.15	1.04–4.43		1 ^c^		
6 months		1.83	0.91–3.72		1.02	0.59–1.78	
18 months		1.49	0.73–3.03		0.62	0.33–1.20	
**Salutogenic Health Indicator Scale (SHIS)**							0.26
Baseline	69.7		67.1–72.4	65.0		61.2–68.8	
6 months	67.1		64.2–70.0	**59.0 ^b^**		**55.0–63.1**	
18 months	67.7		64.7–70.6	**58.9 ^b^**		**54.4–63.3**	
**Cognitive Stress (COPSOQ)**							0.47
Baseline	2.03		1.92–2.14	2.27		2.11–2.42	
6 months	**2.23 ^a^**		**2.11–2.35**	**2.49 ^b^**		2.32–2.66	
18 months	2.14		**2.01–2.27**	**2.52 ^b^**		2.32–2.71	
**Neck/Shoulder Pain**							0.39
Baseline		0.47	0.28–0.81		1 ^c^		
6 months		0.74	0.44–1.23		1.17	0.82–1.68	
18 months		0.69	0.40–1.18		0.99	0.62–1.62	
**Back Pain**							0.73
Baseline		0.55	0.31–0.97		1 ^c^		
6 months		0.69	0.39–1.21		0.99	0.63–1.57	
18 months		0.65	0.36–1.18		1.05	0.64–1.72	

^a^ *p*-value < 0.01, ^b^ *p*-value *<* 0.05, ^c^  reference.

**Table 4 ijerph-18-07640-t004:** Perceived productivity in the activity-based flex office (AFO) in relation to age, gender and work characteristics presented as estimated means (EM). Change in productivity over time is presented as time interaction effect.

	Number, n (%)	Baseline (EM)	6 Months (EM)	18 Months (EM)	*p*-Value for Time Interaction Effect
**Age (y)**					0.935
18–39	53 (27.9)	4.07	3.63	3.56	
40–49	50 (26.3	3.99	3.58	3.55	
50–59	50 (26.3)	3.80	3.33	3.30	
60–	37 (19.5)	4.00	3.43	3.46	
**Gender**					0.778
Female	129 (67.9)	3.95	3.50	3.48	
Male	61 (32.1)	4.00	3.49	3.45	
**Individual Work Tasks Requiring Concentration (h/week)**					<0.001
0–5	43 (22.6)	3.89	3.82	3.79	
6–10	42 (22.1)	3.93	3.53	3.47	
11–15	22 (11.6)	3.89	3.15	3.18	
16–	36 (19.0)	4.07	3.11	3.11	
**Teamwork, >3 Persons (h/week)**					0.011
0–5	101 (53.2)	3.93	3.38	3.39	
6–	29 (15.3)	3.87	3.77	3.74	
**“Spontaneous Meetings Are Important for** **My Work”**					<0.001
1 “I agree”	35 (18.4)	3.99	3.84	3.85	
2	43 (22.6)	3.79	3.52	3.55	
3	41 (21.6)	4.07	3.35	3.30	
4 + 5 = “I disagree”	31 (16.3)	4.00	3.02	2.96	
**Manager**					<0.001
Yes	51 (26.8)	3.85	3.73	3.75	
No	139 (73.2)	4.01	3.42	3.37	
**Computer Related Tasks (h/day)**					0.128
0–4	44 (23.1)	3.97	3.71	3.62	
4–6	69 (36.3)	3.89	3.43	3.45	
6–8	77 (40.5)	4.03	3.44	3.40	
**Individual Work, Routine Tasks (h/week)**					0.197
0–5	65 (34.2)	3.89	3.40	3.31	
6–10	41 (21.6)	3.96	3.47	3.47	
11–	33 (17.4)	3.95	3.62	3.68	
**Hours of Work Involving Phone Calls (h/week)**					0.534
0–5	121 (63.7)	3.93	3.47	3.45	
6–	16 (8.4)	3.87	3.19	3.18	

## Data Availability

The full data are not publicly available due to ethical/privacy reasons. Anonymous data are available for scientific purposes on request from the corresponding author.

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
