# Peer review of "Productivity, Satisfaction, Work Environment and Health after Relocation to an Activity-Based Flex Office—The Active Office Design Study"

_ijerph, 2021, doi:10.3390/ijerph18147640_

Round 1

Reviewer 1 Report

Thank you very much for inviting me to review the mansucript entitled „Productivity, satisfaction, work environment and health after relocation to an activity based flex office“.

The study examines an important intervention using a longitudinal NRCT-design – how a change in work environment affects employee outcomes - with highest relevance and potential interest for readers of IJERPH. Before a publication is possible some minor changes should be conducted by the authors within a revison of the manuscript.

  1. Please have the manuscript proofread by a native speaker before resubmitting, there are some minor but cumulated problems with English grammar.
  2. Please remove double space characters from the manuscript.
  3. You often use slashes in bulleted lists. Please replace them with brackets.
  4. p.5, l.207: I do not finde these numbers (190 and 97) in Figure 1.
  5. For statisfaction you do not report the scale values considered as high or low. I would like to recommend using this a metric measure (which is typical for 1-item-satisfaction-measures). This would also have the benefit of putting analyses in parallel to the other outcomes (reporting odds ratios is a bit confusing here). If you decide not to follow this recommendation please describe how ‚high‘ and ‚low‘ levels of satisfaction (this does also concern the measure in Table 3) were defined.
  6. p.9. Why do you only consider interaction effects in the AFO condition? Similar pattern could have developed in the cell-office. Therefore, you have to show a significant Condition x Moderator x Time –Effect.
  7. You show that there was a selection bias which produced baseline differences in some variables. You might think about using a ‚matching approach‘ (matching persons with similar working conditions and health conditions a baseline) to get the groups more similar and in order to identify the ‚actual‘ intervention effect. Prospensity score analysis might be such an approach.

References:

Austin, P. C. (2020). Advances in propensity score analysis. Statistical Methods in Medical Research, 29(3), 641–643. https://doi.org/10.1177/0962280219899248

Qin, R., Titler, M. G., Shever, L. L., & Kim, T. (2008). Estimating effects of nursing intervention via propensity score analysis. Nursing research, 57(6), 444–452. https://doi.org/10.1097/NNR.0b013e31818c66f6

Reviewer 2 Report

First of all, congratulate the authors for the work done, however, there are certain aspects that need to be improved. So, I list them below:

  1. Regarding the introduction, a theoretical justification is missing that relates the Perceived productivity in the activity based flex-office (AFO) in relation to age and gender, thus, to be able to compare in the discussion.
  2. Regarding the method, you are advised to add a section of participants, where you briefly describe the sample of the same, the distribution by sex, that is, the number of women and men that make up the sample and the mean, as well like age, this should also be included in the summary.
  3. The instruments do not show the coefficient to measure the reliability of the scales used (Cronbach's Alpha), which is why I ask you to include it for each of the instruments used.
  4. Regarding the discussion, Perceived productivity in the activity based flex-office (AFO) in relation to age, gender and work characteristics is not discussed.
  5. Regarding the conclusions section, it is suggested that they include future lines of research.
  6. Regarding the references section, you must review and adapt the references to the journal regulations, since, as they do not contain the numbers of the citations in the references, it makes it difficult to review it.
